

# Fractional solubility of iron in mineral dust aerosols over coastal Namibia: a link with marine biogenic emissions?

**Karine Desboeufs[1], Paola Formenti[1], Raquel Torres-Sánchez[2,3], Kerstin Schepanski[4,$], Jean-Pierre Chaboureau[5], Hendrik Andersen[6,7], Jan Cermak[6,7], Stefanie Feuerstein[4], Benoit Laurent[1], Danitza Klopper[8,*], Aandreas Namwoonde[9], Mathieu Cazaunau[2], Servanne Chevaillier[2], Anaïs Feron[1,%], Cécile Mirande-Bret[1], Sylvain Triquet[1], and Stuart J. Piketh[8]**

[1] Université Paris Cité and Université Paris Est Créteil, CNRS, LISA, F-75013 Paris, France

[2] Université Paris Est Créteil and Université Paris Cité, CNRS, LISA, F-94010 Créteil, France

[3] CIQSO, Robert H. Grubbs Building, University of Huelva, Campus El Carmen, E21071 Huelva, Spain

[4] TROPOS, Leipzig, Germany

[5] Laboratoire d'Aérologie (LAERO), Université de Toulouse, CNRS, UT3, IRD Toulouse, France

[6] Institute of Meteorology and Climate Research, Karlsruhe Institute of Technology (KIT), Karlsruhe, Germany

[7] Institute of Photogrammetry and Remote Sensing, Karlsruhe Institute of Technology (KIT), Karlsruhe, Germany

[8] North-West University, School for Geo- and Spatial Sciences, Potchefstroom, South Africa

[9] SANUMARC, University of Namibia, Henties Bay, Namibia

[*] Now at University of Limpopo, Department of Geography and Environmental Studies, Sovenga, South Africa

[$] Now Institute of Meteorology, Freie Universität Berlin, Berlin, Germany

[%] Now at Université Paris-Saclay, INRAE, AgroParisTech, UMR ECOSYS, Palaiseau, France

**Corresponding author:** paola.formenti@lisa.ipsl.fr





**Abstract**

Mineral dust is the largest contributor to elemental iron in the atmosphere, and, by deposition, to the oceans, where elemental iron is the main limiting nutrient. Southern Africa is an important source at the regional scale, and for the Southern Ocean, however limited knowledge is currently available about the fractional solubility of iron from those sources, as well as on the atmospheric processes conditioning its dissolution during deposition.

This paper presents the first investigation of the solubility of iron in mineral dust aerosols from 176 filter samples collected at the Henties Bay Aerosol Observatory (HBAO), in Namibia, from April to December 2017. During the study period, 10 intense dust events occurred. Elemental iron reached peak concentrations as high as 1.5 µg m$^{-3}$, significantly higher than background levels. These events are attributed to wind erosion of natural soils from the surrounding gravel plains of the Namib desert. The composition of the sampled dust is found to be overall similar to that of aerosols from northern Africa, but characterised by persistent and high concentrations of fluorine, which are attributed to fugitive dust from mining activities and soil labouring for construction.

The fractional solubility of Fe (%SFe) for both the identified dust episodes and background conditions ranged between 1.3 to 20 %, in the range of values previously observed in the remote Southern Ocean. Even in background conditions, the iron fractional solubility was correlated to aluminium and silicon solubility. The solubility was lower between June and August, and increased from September onwards, during the austral spring months. The relation with measured concentrations of particulate MSA (methanesulfonic acid), solar irradiance and wind speed suggests a possible two-way interaction whereby marine biogenic emissions from the coastal Benguela upwelling to the atmosphere would increase the solubility of iron-bearing dust, according to the photo-reduction processes proposed by Johansen and Key (2006). The subsequent deposition of soluble iron could act to further enhance marine biogenic emissions. This first investigation points to the west coast of southern Africa as a complex and dynamic environment with multiple processes and active exchanges between the atmosphere and the Atlantic Ocean, requiring further research.

**Keywords**: aerosols, mineral dust, water-soluble Fe, atmospheric processing, marine biogenic emissions



## 1. Introduction

Through the processes of atmospheric transport and deposition, mineral dust is known to provide nutrients and metals to the terrestrial and marine ecosystems (Hooper et al., 2019; Ventura et al., 2021). Amongst those, mineral dust provides iron (Jickells et al., 2005), which plays a major role for the primary productivity of the nutrient-limited oceans, modulating the marine carbon cycle (Hooper et al., 2019) as well as that of key continental ecosystems such as the Amazon rainforest (Reichholf, 1986).

To date, much attention has been paid to the soluble Fe in mineral dust emitted from arid and semi-arid areas in the northern Hemisphere, in particular the Saharan and Chinese deserts (e.g. Baker et al., 2006; Paris et al., 2010; Takahashi et al., 2011; Rodriguez et al., 2021), where emissions are the most intense (Tegen and Schepanski, 2009).

Nonetheless, the southern Hemisphere accounts for approximately 10% of the global atmospheric dust loading (Kok et al., 2017). Large sources are found in southern Africa, mostly in Namibia (Kala-hari and Namib deserts, Etosha Pan), numerous ephemeral riverbeds along the Namibian coastline) and Botswana (Makgadikgadi Pan; Prospero et al., 2002; Bryant et al., 2007; Mahowald et al., 2003; Ginoux et al., 2012; Vickery and Eckardt, 2013; Von Holdt et al., 2017).

Previous research has shown that the long-range transport of dust emitted from southern African sources can reach the south-eastern Atlantic and the Indian Oceans (Swap et al., 1996; Jickells et al., 2005; Bhattachan et al., 2012; 2015; Ito and Kok, 2017). In particular, Gili et al. (2022) demon-strated recently that mineral dust from Namibia can be transported across the Southern Ocean to eastern Antarctica. Furthermore, the research by Dansie et al. (2022) has suggested that mineral dust from Namibia could dominate the atmospheric deposition to the coastal Benguela Upwelling System (BUS), where biomass burning aerosols, a significant source of soluble Fe to the Southern and Indian Oceans (Hamilton et al., 2021; Ito et al., 2021; Liu, et al., 2022), are limited by atmospheric stratifica-tion (Formenti et al., 2019; Redemann et al., 2021).

There is, however, very little data available on the concentrations and composition of soluble Fe in dust aerosols from southern Africa, both near the sources and over the oceans. Previous research in Namibia focussed on soils and sediments (Dansie et al., 2017a; 2017b; Kangueehi, 2021). The At-lantic Meridional Transect (AMT) cruise programme conducted recurrent observations between Oc-tober and March in the South Atlantic Ocean (Baker et al., 2013), while Heimburger et al. (2013) and Gao et al. (2013) report on sparse measurements of deposited aerosols and in rainwaters over the Southern Indian Ocean.

Within this context, this paper investigates the fractional solubility of Fe in samples of atmospheric aerosol particles smaller than 10 µm in diameter collected in 2017 at the Henties Bay Aerosols Ob-servatory (HBAO; 22.09°S, 14.26°E) on the Namibian coast. In section 2 we outline the experimental



and analytical methodology for elemental and water-soluble analysis of ions and metals, including
iron, obtained by Inductively Coupled Plasma (ICP) analysis. We also provide the definition of frac-
tional solubility and method for estimating the total dust mass. We introduce the supporting tools used
to evaluate the source regions of the collected mineral dust, their pathways during transport, and the
presence of fog, a recurrent feature on coastal Namibia favouring multi-phase ageing processes.
Section 3 provides the results of the analysis. We present the iron soluble concentrations and solu-
bility, and explore their links to the load, emission area and transport of mineral dust, as well as at-
mospheric processing. Section 4 discusses the observations, suggesting that the fractional solubility
of iron in the Namibian dust is higher when the particulate MSA, a tracer of marine biogenic emissions,
is also detected in highest concentrations. This points to the photo-oxidation of DMS as a process for
increasing the dust solubility, and suggests a possible positive feedback loop of the iron fertilisation
by dust to the ocean. Section 5 summarizes the findings and suggests directions for future research.

## 2. Methodology

### 2.1. Study area

The Henties Bay Aerosol Observatory (HBAO, 22.09°S, 14.26°E; http://www.hbao.cnrs.fr/, last ac-
cess: 10 October 2022) is located at the Sam Nujoma Marine and Coastal Resources Research Cen-
tre (SANUMARC) of the University of Namibia in Henties Bay, Namibia (**Fig 1**).

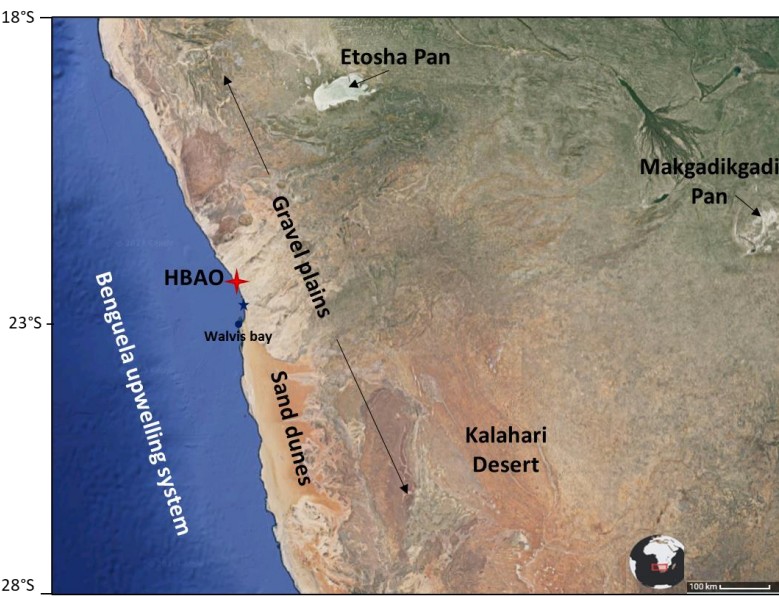

**Fig 1.** *Location of Henties Bay Aerosol Observatory (HBAO, red star) and main dust source regions (© Google Maps). The position of Walvis Bay (blue dot), the major harbour in the area, and the Wlotzkasbaken meteorological station (blue star) are also indicated.*




Three kilometers to the south of the University campus hosting HBAO is the small town of Henties
Bay, with no industrial activity and very little traffic, and approximately 170 km north from Walvis Bay,
the major harbour in Namibia. Directly east of HBAO are the Namibian gravel plains, which are one
of the dominant features of the Namib desert together with the sand dunes. Approximately 100 m to
the north is the Omaruru riverbed, one of the coastal sources of mineral dust identified by Vickery and
Eckardt (2013).
Our previous results show that, at the surface level, the atmosphere at HBAO is a receptor of different
air masses dominated by marine aerosols, but also the seasonal occurrence of light-absorbing aero-
sols from biomass burning or pollution in northern wind regimes, and mineral dust detected episodi-
cally from various wind directions (Formenti et al., 2018; Klopper et al., 2020, hereafter KL20).
**2.2. Sample collection and analysis**
Aerosol particles smaller than 10 μm in aerodynamic diameter (PM$_{10}$) were collected by an automated
sampler (model Partisol Plus 2025i, Thermo Fisher Scientific, Waltham, MA USA) on 47 mm What-
man Nuclepore polycarbonate filters (1-μm pore size). The air was drawn through a certified sampling
inlet (Rupprecht and Patashnick, Albany, New York, USA) located at approximately 30 m above
ground and operated at a flow rate of 1 m$^3$ h$^{-1}$. Samples were collected for 9 hours during the daytime
(from 9:00 to 18:00 UTC time) and night-time (21:00 to 06:00 UTC time) for 12 non-consecutive weeks
from April to December 2017 (7-14 April, 26 April-3 May, 19-26 May, 07-14 July, 2-9 August, 15-22
August, 18-25 September, 02-09 October, 31 October-7 November, 13-20 November, 28 November-
04 December, 12-19 December). In total, 176 samples (including 13 blanks, one per week of sam-
pling) were collected.
The elemental analysis of 24 elements from Na to Pb and including some major tracers of mineral
dust (Fe, Al and Si) was performed at the LISA laboratory by Wavelength-dispersive X-ray fluores-
cence (WD-XRF) using a PW-2404 spectrometer (Panalytical, Almelo, Netherlands), as detailed by
KL20. The total mass concentration per element x will be referred to as Tx.
The measured elemental concentrations are used to calculate the estimated dust mass (*EDM*) ac-
cording to Lide (1992) as
*EDM* = 1.12 x {1.658 x [nss-Mg] + 1.889 x[Al] + 2.139 x [Si] + 1.399 x [nss-Ca] + 1.668 x [Ti] + 1.582
x [Mn] + (0.5 x 1.286 + 0.5 x 1.429 + 0.47 x 1.204) x [Fe]}    (1)

where, as explained by KL20, nss-Mg and nss-Ca represent the non-sea salt fractions of Mg and Ca,
respectively.



The analysis of the water-soluble fraction was also performed at LISA. Individual filters were placed
in 20 mL of ultrapure water (MilliQ® *18.2* MΩ.cm) for 30 minutes. The solution was then divided into
two sub-samples. One half was analysed by Ion chromatography (IC) using a Metrohm IC 850 device
equipped with a column MetrosepA supp 7 (250/4.0 mm) for anions and with a Metrosep C4 (250/4.0
mm) for cations. The IC analysis provided the concentrations of the following water-soluble ions: for-
mate, acetate, MSA⁻ (methanesulfonic acid), $Cl^-$, $NO_3^-$, $SO_4^{2-}$, oxalate, $Na^+$, $NH_4^+$, $K^+$, $Ca^{2+}$ and $Mg^{2+}$.
A calibration with certified standard multi-ions solutions of concentrations ranging from 5 to 5000 ppb
was performed and the uncertainty of the analysis was estimated to be 5% (KL20).
The second half of the solution was acidified to 1% with ultrapure nitric acid ($HNO_3$) and analysed by
Inductively Coupled Plasma-Atomic Emission Spectroscopy (ICP-AES) using a Spectro ARCOS
Ametek® ICP-AES and by High-resolution Inductively Coupled Plasma-Mass Spectrometry (HR-ICP-
MS) using a Neptune Plus™ instrument by Thermo Scientific™. The calibration curve was performed
using standard multi-element solutions ranging from 2 to 1000 ppb for ICP-AES and 1 to 1000 ppt for
HR-ICP-MS (Desboeufs et al., 2022). These analyses provided the dissolved mass concentrations
(Dx) of 25 water-soluble metals and metalloids, including Fe, Al, and Si. All sample concentrations
were corrected using the filter blanks for each sampling period.
Based on those analyses, the fractional solubility (%Sx) representing the percentage solubility value
was calculated as

$$\%S_x = 100 \times D_x/T_x \qquad (2)$$

with Dx and Tx, the dissolved and total elemental concentration respectively.
**2.3.  Ancillary data**
Maps of the emission fluxes of mineral dust were calculated using the dust emission model described
by Feuerstein and Schepanski (2019), driven with hourly 10m wind fields at a 0.1° x 0.1° grid from
the European Centre for Medium-range Weather Forecasts (ECMWF). The dust emission parame-
terisation follows Marticorena and Bergametti (1995). Additional information on the soil type was taken
from the ISRC soil data set (FAO/IIASA/ISRIC/ISSCAS/JRC, 2012) and information on the aerody-
namic roughness length was obtained from POLDER/ADEOS surface products following the works
of Marticorena et al. (2004) and Laurent et al. (2005). The MODIS monthly vegetation product
(MYD13A3 v6) was used to describe the vegetation cover, while the vegetation type was defined
using the BIOME4 database (Kaplan et al., 2003). We additionally differentiated between different
dust source types (alluvial fines, dunes and sand sheets) which allowed us to reflect the source di-
versity over Namibia and thus the spatial diversity in the soil's susceptibility to wind erosion. This layer



was compiled following Feuerstein and Schepanski (2019) using MODIS surface reflectance
(MOD09A1 v6). A MODIS retrieved map on surface water cover was used to eliminate flooded areas
as active dust sources.
Back-trajectories of the air masses during the dust event were calculated from Meso-NH model (ver-
sion 5.3). The model set-up is similar to the one used for the AErosols, RadiatiOn and CLOuds in
southern Africa (AEROCLO-sA) field campaign (Formenti et al. 2019) and related case studies (Fla-
mant et al. 2022; Chaboureau et al. 2022). In short, the model was run on a 5 km grid covering the
southern tip of Africa and 67 stretched levels spaced by 60 m close to the surface and 600 m at high
altitude. Meso-NH was run for 24 h for each dust event using initial and boundary conditions provided
by the ECMWF operational analysis. Emission, transport and deposition of dust is described by the
scheme of Grini et al. (2006). Back trajectories were computed online using three passive tracers
initialized with the 3D-field of their initial conditions. Further details on the dust prognostic scheme,
the backward trajectories and the physical parameterizations are given in Chaboureau et al. (2022).
The presence of fog and low clouds (FLC) along the Namibian coastline during dust events was an-
alysed using an existing satellite-based fog and low-cloud data set (Andersen et al., 2019). The FLC
detection algorithm used to create this data set was developed and validated specifically for this re-
gion. The algorithm is based on infrared observations from the Spinning Enhanced Visible and Infra-
red Imager (SEVIRI) aboard the geostationary Meteosat Second Generation (MSG) satellites, making
use of both spectral and textural information. The FLC product is available at the native spatial and
temporal resolutions of the SEVIRI sensor (3 km nadir, every 15 minutes), as described in Andersen
and Cermak (2018). The FLC product does not specifically distinguish between fog and low clouds
but captures the coastal boundary-layer cloud regime typical for the region and at HBAO that could
interact with mineral dust. It has been shown to be consistent with synoptic-scale atmospheric dy-
namics (Andersen et al. 2020). The FLC data are used to calculate maps of average fog and low
cloud coverage for the time periods of all dust events given in Table 1.
Observations of the local meteorology, including measurements of air temperature, relative humidity
and fog, at the nearby Wlotzkasbaken meteorological station (22.31°S, 14.45°E, 73 m asl, see **Fig.**
**1**) part of the Southern African Science Service Centre for Climate Change and Adaptive Land Man-
agement (SASSCAL) ObservationNet (https://www.sasscal.org/; last accessed 14/04/2023), are
used.

**3. Results**
**3.1.    Description of the dust episodes**

The dataset discussed in this paper is based on 176 aerosol samples collected at HBAO, 42 of which were associated with 10 dust episodes. As detailed by KL20, events of mineral dust were identified as peaks in the time series of the mass concentrations of Al and non-sea-salt $Ca^{2+}$ (nss-$Ca^{2+}$). The dust episodes investigated in this study are a subset of those presented by KL20, we therefore use their naming convention to facilitate the connections between the two papers (**Table 1**). In the following, we refer to samples collected during the dust episodes as “dust”. Samples collected outside the dust events will be indicated as “background”.

**Table 1.** *Dates of dust events identified at HBAO from May to December 2017, following KL20. The number of samples collected during each episode is indicated in the column called “N”. The average air temperature, relative humidity, wind speed and direction recorded at the nearby meteorological station in Wlotzkasbaken are reported. The maxima wind speed and corresponding direction are indicated in brackets in the corresponding columns. The average EDM is reported with in brackets the maximum of EDM during the event.*

| Episode identifier | Start and end date (UTC) | N | Air temperature (°C) | RH (%) | Wind speed ($m\ s^{-1}$) | Wind direction (degN) | EDM ($\mu g\ m^{-3}$) |
|---|---|---|---|---|---|---|---|
| **Dust 04** | 19/05 09h – 20/05 18h | 3 | 17.7 | 73.7 | 2.7 (6.2) | 186 (185) | 13 (14) |
| **Dust 05** | 24/05 21h – 26/05 09h | 3 | 18.1 | 63.3 | 2.3 (6.3) | 183 (188) | 21 (42) |
| **Dust 06** | 11/07 09h – 13/07 09h | 4 | 13.2 | 82.9 | 1.2 (5.4) | 235 (193) | 27 (45) |
| **Dust 07** | 04/08 21h – 06/08 09h | 4 | 12.5 | 87.0 | 1.2 (5.4) | 233 (201) | 10 (16) |
| **Dust 08** | 17/08 21h – 19/08 09 h | 4 | 11.9 | 80.6 | 1.3 (4.6) | 324(129) | 18 (21) |
| **Dust 09** | 23/09 21h – 24/09 18h | 2 | 15.6 | 84.3 | 3.1 (6.2) | 309 (330) | 11 (17) |
| **Dust 10** | 05/10 21h – 08/10 09h | 8 | 14.0 | 74.6 | 2.1 (5.9) | 249 (228) | 14 (23) |
| **Dust 11** | 15/11 09h – 18/11 09h | 6 | 16.7 | 66.1 | 3.2 (11.7) | 231 (232) | 31 (56) |
| **Dust 12** | 30/11 09h – 01/12 18h | 3 | 16.7 | 78.1 | 1.9 (5.7) | 244 (195) | 2 (3) |
| **Dust 13** | 15/12 09h – 19/12 09h | 7 | 16.9 | 76.9 | 2.9 (6.5) | 252 (238) | 10 (19) |

The dust episodes were long-lasting (generally a few days). The dynamic of the emissive areas, air mass transport and fog coverage during the episodes (**Fig. S1**) is driven by the synoptic circulation, which, in southern Africa, is primarily affected by the high-pressure belt under the descending limb of the Hadley cell (Tyson and Preston-Whyte, 2014). The maps of dust emission fluxes and the air mass back-trajectories reflect this seasonality. During the first part of the year (episodes Dust 04 to 05), dust emissions originated from the gravel plains and the Etosha pan north of HBAO. During this time of the year the transport to HBAO below 300 m asl was north- to south-easterly originating inland from the coast.

From July onwards, the active source areas were identified in the southern gravel plains, Namib sand dunes and Kalahari Desert (this former only for Dust 11 to 13). Air mass transport was southerly and travelled over the sea and along the coastline. It is worth noticing that all the air masses experienced





maritime air during their last hours of transport, including the episodes Dust 04 and 05 associated
with berg wind conditions, due to the coastal low that develops to the west of HBAO.
The formation of fog events at Henties Bay is also highly seasonal. The frequency of occurrence of
fog events is highest during austral winter at the coast, whereas lifted stratus clouds dominate during
austral summer, when overall FLC occurrence peaks. The occurrence of fog over Namibia correspond
to the advection of low-level clouds which is modulated both by local meteorology along the coastline
of Namibia and synoptic-scale radiative processes (Spirig et al., 2019; Andersen et al., 2019; 2020).
Henceforth, as shown in Fig. S1, the presence of fog and low clouds correlates with wind directions
and aerosol source regions. Overall, three episodes (Dust 04, Dust 05 and Dust 11 in April, May and
November, respectively) occurred in fog-free or low-fog conditions. The remaining episodes were
characterised by extensive fog and low cloud coverage throughout the study area. The meteorological
observations at the nearby Wlotzkasbaken station (**Fig. S2**) confirm these findings, and show in par-
ticular that the relative humidity always exceeded 60 %, and 80 % when fog or low clouds were pre-
sent (Table 1). As a consequence, the aerosol can be considered deliquescent even in the fog-free
conditions. The seasonality is also observed in the average downwelling solar irradiance, with the
lowest values during July and September, associated with austral winter. Finally, it is interesting to
note that the fog-free conditions, associated with the predominance of continental air masses, corre-
sponded to the highest estimated dust mass (EDM), possibly because of the reduced wet removal
during transport and the increase of emission fluxes with the decrease of soil moisture (Kok et al.,
2014), but possibly also because of the high wind speed prevailing during these conditions, which in
principle, enhancing both dust emissions and transport (Table 1).
**3.2.    Iron solubility**
The total and dissolved concentrations, and fractional solubility of Fe, Al and Si, during the dust epi-
sodes are reported in **Table 2,** where they are compared to background conditions. For iron, the
average values over the entire sampling period are also shown.
*Table 2. Average and standard deviations of water-soluble (Dx), total elemental (Tx) mass concentrations and*
*fractional solubility (%Sx) for Fe, Al and Si at HBAO measured for the total period and during the dust and*
*background events from April to December 2017. Concentrations values are expressed in ng m$^{-3}$, while frac-*
*tional solubility is expressed in percent. The numbers of considered samples is presented between the paren-*
*theses.*

|  | Fe | | | Al | | Si | |
|---|---|---|---|---|---|---|---|
|  | **All period** | **Dust** | **Background** | **Dust** | **Background** | **Dust** | **Background** |
| **Dx** | 28 ± 51 *(N=175)* | 80 ± 84 *(N=42)* | 11 ± 10 *(N=131)* | 322 ± 296 *(N=42)* | 56 ± 46 *(N=131)* | 529 ± 616 *(N=42)* | 78 ± 83 *(N=124)* |
| **Tx** | 364 ± 482 *(N=176)* | 955 ± 633 *(N=42)* | 177 ± 155 *(N=133)* | 1204 ± 870 *(N=42)* | 284 ± 222 *(N=94)* | 4158 ± 3037 *(N=42)* | 776 ± 674 *(N=133)* |
| **%Sx** | 7.1 ± 3.6 *(N=175)* | 7.9 ± 4.1 *(N=42)* | 6.8 ± 3.3 *(N=130)* | 27 ± 10 *(N=42)* | 26 ± 11 *(N=90)* | 12 ± 7 *(N=42)* | 11 ± 8 *(N=116)* |




The total Fe concentrations varied significantly from one episode to the other, and so did EDM, which
was larger than 10 µg m$^{-3}$ for all of them (except Dust 12) and as high as 56 µg m$^{-3}$ during Dust 11
event (Table 1). By contrast, the total Fe-to-EDM ratio was virtually constant, with an average of 5.8
% (± 0.6 %) for the dust events and 5.6 % (± 1.1%) for the entire dataset.
The total dissolved concentrations of Fe during the sampling period ranged from 1.5 to 427 ng m$^{-3}$,
with a median and average of 10.5 and 28 ng m$^{-3}$. During the dust episodes, the average mass con-
centration of dissolved Fe was 80 ± 84 ng m$^{-3}$, almost an order of magnitude higher than for back-
ground conditions (11 ± 10 ng m$^{-3}$). The dissolved concentrations in dust periods are higher than
those observed in the South Atlantic Ocean for air masses associated with transport from continental
southern Africa (Baker et al., 2013; Chance et al., 2015; Baker and Jickells, 2017), which are of the
order as those observed at HBAO for background periods.
The calculated fractional solubility of Fe ranged from 1.3 to 19.8 %, with a median and average of 6.7
and 7.1 %. The average %SFe during dust events (7.9 ± 4.1%) was higher than in background con-
ditions (6.8 ± 3.3%). It is interesting to note that Dust 11 event, the most intense recorded event,
presents the highest %SFe (between 10.2 and 19.8 % with an average at 13.8 %). Apart from this
event, the average fractional solubility seems to be independent of the EDM. Excluding this event, the
average solubility of Fe for dust event (6.9 % ± 3.3 %) is equivalent to the one for background samples.
For both conditions, the observed range of variability is high and consistent with previous observations
(2.4-20 %, Baker et al., 2013; 1.3-22 %, Chance et al., 2015), as well as with measurements over the
Southern Indian Ocean (0.76-27 %, Gao et al., 2013).
The temporal variability of %SFe is presented in **Fig. 2**, where dust and background episodes are
shown separately.

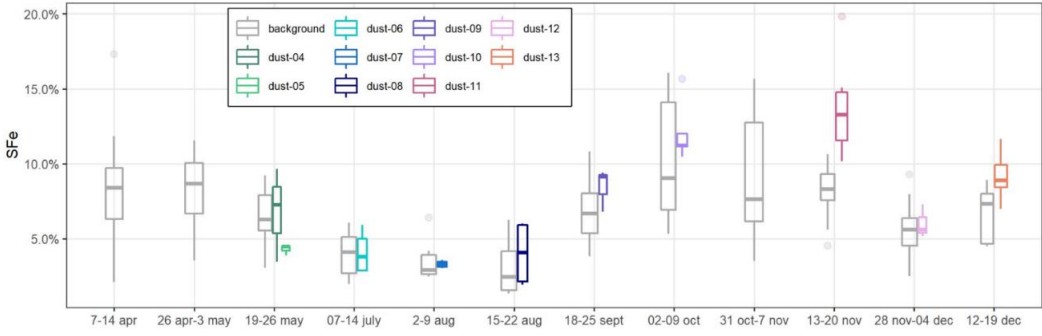


**Fig.2**: *Temporal variability of %SFe average for dust and background samples during the different periods of*
*sampling. In the box plots, the box indicates the interquartile range, i.e. the 25th and the 75th percentile, and*
*the line within the box marks the median. The whiskers indicate the quartiles ±1.5 times the interquartile range.*
*Points above and below the whiskers indicate outliers outside the 10th and 90th percentile.*






The temporal variability is similar during dust and background conditions. The highest %SFe occurred
during austral spring (October-November), and in particular during episode Dust 11 from 13 to 20
November 2017, when the average %SFe reached 13.8 %. The %SFe was quite similar along the
year between dust and background, except between 13-20 November where the iron solubilities dur-
ing Dust 11 event was very superior to the one of background samples, and to a lesser extent, in
September (Dust 09) and December (Dust 13).
**Fig. 3** represents the correlations of Fe with Al and Si, both for the total and the dissolved concentra-
tions.

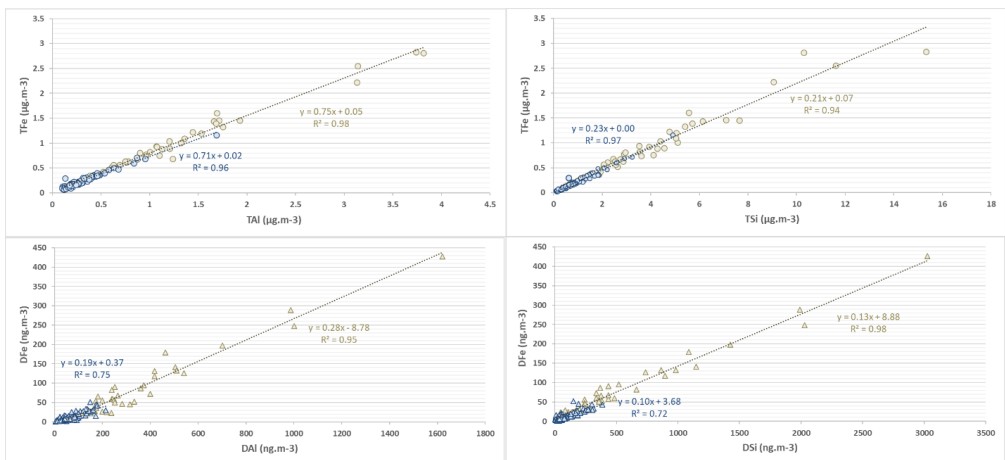


**Fig. 3.** *Scatterplot of TFe with respect to TAl and TSi (top panels) and DFe with respect to DAl and DSi (bot-*
*tom panels) for dust (sand dots and triangles) and background events (blue dots and triangles). The Pearson*
*coefficient are shown for both.*

For both dust and background samples, the total Fe concentration is linearly correlated with total Al
($R^2$=0.98 and 0.96, slope=0.75 and 0.71, for dust and background conditions respectively) and total
Si ($R^2$=0.94 and 0.97, slope=0.21 and 0.23, respectively). The slopes are consistent with typical Fe/Al
and Fe/Si ratios found in desert dust from northern Africa (Formenti et al., 2014; Shelley et al., 2014),
confirming the main crustal origin of Fe during all the sampling periods. Likewise, the concentrations
of dissolved iron (DFe) show a strong linear correlation with both DAl and DSi, for both for dust and
background events ($R^2$=0.96 and 0.75 with respect to DAl and $R^2$=0.98 and 0.73 with respect to DSi).
The slopes for Al and Si are also comparable (0.19 and 0.28 for DAl and 0.10 and 0.13 for DSi,
respectively in dust and in background events). A very strong linear correlation was also observed
between DFe and DTi ($R^2$=0.96 and 0.84; not shown), another unique marker of mineral dust. Signif-
icant correlations of soluble concentrations for several elements associated with mineral dust (Fe, Al,





Si, Ti) have been previously obtained in remote aerosols over ocean area (Baker et al., 2016). Addi-
tionally, DFe during dust events correlate very closely with F$^-$ (R$^2$=0.94, not shown), which has been
indicated by KL20 as being emitted in the atmosphere by the wind erosion as well as the labouring of
the Namibia soil, rich in fluoride mineral deposits.

## 4. Discussion

Several studies have showed that variations in aerosol Fe solubility could result from the source/com-
position of the aerosols. As a matter of fact, the Fe solubility has been linked to the iron mineralogy
(Journet et al., 2008) and has been shown being lower for African crustal sources than in continen-
tal/anthropogenic sources (Desboeufs et al., 2005; Sholkovitz et al., 2009; Shelley et al., 2018). The
iron fractional solubility in mineral dust is also affected by source mixing (Paris et al., 2010; Desboeufs
et al., 2005), by (photo)chemical processing with acids or organic ligands during atmospheric
transport (Paris et al., 2011, Paris et Desboeufs, 2013; Wozniak et al., 2013; Swan and Ivey, 2021)
and by the increase of surface area to volume ratio due to size changes during transport (Baker &
Jickells, 2006; Marcotte et al., 2020).
In the following sections, we discuss these possible factors to explain the seasonality and the ex-
tended range of variability of the fraction Fe solubility in HBAO samples. The possible increase of
surface area to volume ratio during transport (Baker and Jickells, 2006; Marcotte et al., 2020) will not
be discussed because of lack of appropriate observations of the size distribution. Because of the
similar transport time suggested by back trajectories (Fig.S1), it is likely that particle size distribution
would be similar from one event to the other.

### 4.1. Influence of dust composition

Close to dust source, iron solubility could be mainly conditioned by the mineralogical composition of
dust (Journet et al., 2008, Formenti et al., 2014). Considering that soluble Fe-bearing aerosols were
issued from mineral dust for all the samples, the seasonality of dust emission sources (see 3.1) could
be a factor explaining the seasonality of %SFe (and other elements associated to mineral dust). **Fig.**
**4** shows the scatter plot of the elemental mass ratio of Fe/nss-Ca$^{2+}$ and Si/Al, previously used for
northern Africa dust to distinguish aerosol dust from source areas enriched in clays or iron oxides to
soils rich in quartz or carbonates (Formenti et al., 2014). Specific to Namibia, because of the strong
link between nss-Ca$^{2+}$ and fluorine, the Fe/nss-Ca$^{2+}$ ratio may also to distinguish dust influenced by
fluorspar mining.

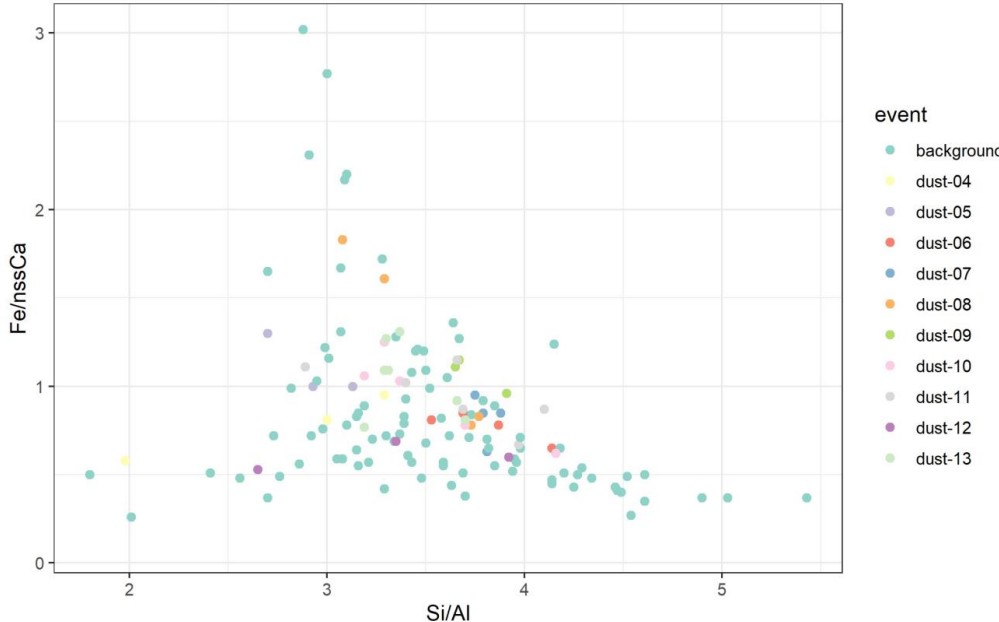

**Fig 4.** *Scatterplot of Fe/nss-Ca$^{2+}$ and Si/Al mass ratios for the samples collected at HBAO in period May-December 2017. Values obtained for samples collected during the dust events are represented as brown dots. Values for samples collected outside those events (background) are represented as blue dots.*

Figure 4 indicates that the range of variability of both Fe/nss-Ca$^{2+}$ and Si/Al ratios is small when considering dust events only. The elemental ratios of samples collected during the background periods are rather similar to dust events during a same sampling period, except for Si/Al for the period between 19-26 May and for Fe/nss-Ca$^{2+}$ for the samples of 18-25 September, when significant differences, not really explicable and not inducing a significant difference in the %SFe values are observed (**Fig. S3**).

The values for ambient dust measured at HBAO are consistent with those of the previous field observations in Namibia (Annegarn et al., 1983; Eltayeb et al., 1993), but also with values reported by Caponi et al. (2017) for laboratory-aerosolised dust from two soils collected on the Namibian gravel plains. This is in agreement of the indications of the emission maps (**Fig. S1**), showing significant emissions in the gravel plains. The absence of seasonal cycle in the elemental composition illustrated in **Fig. S3** suggests that the seasonal change from northern to southern sources does not induce a change in the composition of the aerosol dust sampled at HBAO, which is consistent with the fact that the northern and the southern gravel plains of Namibia have similar mineralogy (Heine and Vökel, 2010). This suggests that the mineralogical composition of mineral dust should not be a discriminating factor explaining the seasonality of the iron solubility observed at HBAO.





**4.2. Evidence of processing by marine biogenic emissions**

The atmospheric (in-cloud) processing associated with secondary aerosol production may increase the fractional solubility of Fe during transport (Takahashi et al., 2011; Rodríguez et al., 2021). This has also been shown for Al and Ti (Baker et al., 2020). The chemical processing could include both acidic and ligand-promoted dissolution (Desboeufs et al., 2001, Longo et al., 2016, Tao et al., 2019). Oxalic acid has previously been used as a proxy for organic ligand–mediated iron dissolution processes because it is the most abundant species in the atmosphere and is the most effective ligand in promoting iron dissolution (Baker et al., 2020; Hamilton et al., 2021). However, several secondary compounds, such as carboxylate ligands and marine secondary products derived from dimethyl sulfide (DMS) oxidation, have been identified as playing a role in increasing the solubility fraction of iron from mineral aerosols (Johansen and Key, 2006; Paris et al., 2011; Paris and Desboeufs, 2013; Wozniak et al., 2013 and 2015). The increase of ligands-promoted dissolution is attributed to photochemical reduction of Fe(III) in Fe (II) (Siefert et al., 1994; Johansen and Key, 2006).

To investigate these aspects, the mass concentrations of the ionic compounds (oxalate, formate, MSA, $NO_3^-$, $NH_4^+$ and $nss-SO_4^{2-}$) implied in the secondary aerosol production, measured at HBAO during dust and background periods are reported in **Table 3**.

**Table 3.** *Average and standard deviations of mass concentrations of water-soluble ions measured at HBAO during dust and background events from May to December 2017. Concentrations are expressed in ng m$^{-3}$. The number of samples pertaining to each occurrence is indicated in brackets.*

|  | **Dust** | **Background** |
|---|---|---|
| $nss-SO_4^{2-}$ | 1795 ± 762 (N = 42) | 1366 ± 505 (N=132) |
| Oxalate | 155 ± 53 (N = 42) | 127 ± 35 (N = 132) |
| Formate | 18 ± 6 (N = 40) | 16 ± 9 (N = 105) |
| MSA | 64 ± 37 (N=36) | 56 ± 36 (N=114) |
| $NO_3^-$ | 205 ± 79 (N=42) | 200 ± 138 (N=132) |
| $NH_4^+$ | 192 ± 71 (N=42) | 207 ± 98 (N=132) |

Oxalate was the most abundant organic compound, followed by MSA, a secondary product of DMS oxidation and a unique particulate tracer of the primary marine biogenic activity (Andreae et al., 1995). On average, organic compounds were equally concentrated in dust and background events. Amongst inorganic species, $nss-SO_4^{2-}$ was the most concentrated compound, with higher values during the dust events than during the background period.

Their detailed time series are shown in **Fig 5**, where it is compared to that of the iron fractional solubility.

**Fig. 5.** *Box-plots of the averages of %SFe and secondary organic and inorganic compounds mass concentrations (μg m⁻³) for the sampling periods including all the samples (dust + background). Boxes and whiskers as in Fig. 2.*






There is no clear seasonal cycle for any of the ionic compounds, with the exception of MSA, which
shows a similar time variability than %SFe. MSA concentrations were lowest between May and Au-
gust (average 38.0 ± 28.0 ng m$^{-3}$), while higher concentrations were measured from September to
December (72.7 ± 38.1 ng m$^{-3}$). These differences are also observed for the dust cases only. The
average MSA concentration was 40.6 ± 23.4 ng m$^{-3}$ for Dust 04 to Dust 08 episodes. It increased to
77.7 ± 35.3 ng m$^{-3}$, almost a factor of 2 between episodes Dust 09 and Dust 13. The mass concen-
trations and the seasonal cycle of MSA are related with the proximity of the strong coastal upwelling
by the Benguela current (Formenti et al., 2019; KL20). The maximum concentration of MSA (106.2
ng m$^{-3}$) was measured during episode Dust 11, which is also the time of the highest SFe% observa-
tion. This episode was also characterised by the highest oxalate, nss-SO$_4^{2-}$ and NO$_3^-$ concentrations.
Based on their temporal variability, Fig. 6 shows the correlation plot between total Fe, Al and Si, and
their respective fractional solubility, the measured secondary compounds and the meteorological con-
ditions during sampling obtained from Principal Component Analysis (PCA) for all the samples. The
variables correlated in time are grouped together (the closer they are to the circle, the stronger the
correlation) whereas the variables which are anti-correlated are situated on the opposite side of the
plot origin.

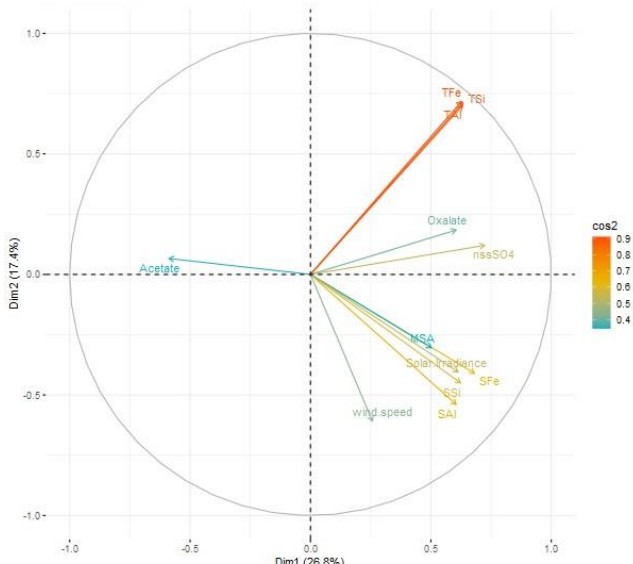

**Fig 6.** *PCA analysis performed from the database including %SX and secondary ions concentrations. The scale(cos2)*
*gives the factor of correlation between the different parameters. Formate, nitrate and ammonium are not visible in the plot*
*showing that they are not correlated with the other parameters.*






The PCA correlation emphasizes the dependence between %SFe (%SAI and %SSi) and the MSA concentrations (correlation factor around 0.4), while indicating a weak dependence on oxalate, acetate and nss-$SO_4^{2-}$. Fig. 6 also shows that %SFe is correlated with both the wind speed and solar irradiance (correlation factor higher than 0.6). While it is expected that the emission of mineral dust occurs when the wind speed is high, the correlation of %SFe with wind speed is rather surprising as both Table 1 and Fig.2 show that the %SFe is independent of the dust load. **Fig. S4** in the supplementary material shows that the wind speed is also correlated with the MSA concentrations. This is consistent with Andreae et al. (1995), who demonstrated how, in this area due to persistent phytoplankton bloom, the atmospheric concentrations of dimethylsulphide (DMS), the gaseous precursors of MSA, depend on the sea-to-air flux, in turn is determined by the concentrations in the ocean water and the surface wind speed. On the other hand, the MSA concentrations do not correlate significantly with the average solar irradiance.

As previously mentioned, Johansen and Key (2006) showed an increase of dissolution of ferrihydrite, a proxy of iron(oxy)hydroxide found in desert mineral dust, by photolysis of the Fe(III)-MSIA (methanesulfinic acid) complex, producing MSA and soluble Fe. Zhuang et al. (1992) proposed an increase of iron dissolution by the acidification of aerosol particles associated with dimethylsulphide (DMS) oxidation. Here, the link between the Fe fractional solubility, solar irradiance and MSA is in agreement with the photo-reduction dissolution of Fe by MSA condensation on Fe-bearing dust. Thus, we attribute the iron fractional solubility seasonality observed at HBAO both to solar irradiance and MSA temporal evolution via this process.

**4.3. Link to other sources of iron and oxalate**

The mass apportionment of iron reported by KL20 indicates that, during the dust events and the background periods, respectively, 7% and 29% of the mass of total elemental Fe was not associated to mineral dust, but rather to a factor indicated as "ammonium-neutralised component", mostly characterised by secondary species, and non-sea-salt potassium (nss-$K^+$). Because of this association, previously reported by Andreae (1983), the "ammonium-neutralised component" was associated to photo-oxidation of marine biogenic emission but also episodically to biomass burning, which can be transported to HBAO during the Austral summertime, when the airflow becomes anti-cyclonic, and the transport of air masses laden with light absorbing aerosols has been documented (Formenti et al., 2018).

However, our data do not indicate any significant dependence of %SFe to the percent mass fraction of iron attributed to sources other than dust, notably combustion particles, which was expected in the light of previous research (e.g. Desboeufs et al., 2005; Sholkovitz et al., 2009; Shelley et al., 2018;



Ito et al., 2021), and indeed the lowest Fe solubility (< 5%) was measured in July and August 2017,
when the contribution of polluted air masses should be highest.
The "ammonium-neutralised component" identified by KL20 included oxalate, the most concentrated
organic species at HBAO, and the strongest of the organic ligands promoting the photo-reduction of
iron in mineral dust, henceforth the increase of its fractional solubility (Paris and Desboeufs, 2013).
Surprisingly, excepted individual cases (Dust 13), our analysis does not show this strong link (Fig. 6),
which we explain by the fact that, contrary to the SFe%, the oxalate concentrations measured at
HBAO was practically constant with time, the possible pathways of oxalate formation in this complex
atmosphere being numerous and occurring through the year, from natural and anthropogenic sources
(marine, heavy-oil combustion, biomass burning) and in-cloud and photo-oxidative processes (Ba-
boukas et al., 2000; Myriokefalitakis et al., 2011).

**5.    Conclusive remarks**
For the first time, the fractional solubility of Fe in airborne atmospheric aerosols smaller than 10 μm
in diameter is investigated along the west coast of Namibia, in southern Africa, a critical region for the
global climate.
Ten intense episodes of transport of mineral dust from aeolian erosion were identified from the anal-
ysis of aerosol samples collected between May and December 2017 at the Henties Bay Aerosol Ob-
servatory (HBAO). Based on modelling and measurements, source regions were identified both in the
northern and southern gravel plans. Our data do not provide any evidence of the possible contribution
of dust from coastal riverbeds, which are considered to be frequent sources of atmospheric dust and
soluble iron in the region. (Vickery et al., 2013; Von Holdt et al., 2017; Dansie et al., 2017a; 2017b).
Our first measurement indicate that the total iron represents, on average, 5.8 % (± 0.6 %) of the total
dust mass, and that the average iron fractional solubility is 6.9 % (± 3.3 %). These values should be
useful to atmospheric models estimating the dust-borne input of soluble Fe from the gravel plains in
Namibia to the surrounding oceans.
The measured iron fractional solubility is comparable to values reported from shipborne measure-
ments of transported dust in the remote southern oceanic regions (Baker et al., 2013; Chance et al.,
2015, Gao et al., 2013) but significantly higher than obtained in a benchmark laboratory evaluation
from the same soils and an identical dissolution protocol (*unpublished data*). The time series of frac-
tional solubility of Fe shows an apparent seasonal cycle which is independent of dust composition.
This is also the case for Al and Si.



The observations presented in this paper exclude a major role of sources other than mineral dust to
play on the values and the variability of %SFe, which might be due to the location of our sampling
site, remote and only occasionally affected by polluted air masses (Formenti et al., 2018).
Conversely, the seasonal increase of the iron fractional solubility is associated to that of the concen-
trations of MSA and correlated to meteorological parameters such as the wind speed and the surface
solar irradiance. Our observations support the role of photo-chemical processes in the dissolution of
Fe in our samples, and suggest that the oxidation of the marine biogenic emissions from the northern
Benguela upwelling, favoured under high wind speed conditions, could play a significant role in in-
creasing the solubility of elemental iron in mineral dust aerosols over coastal Namibian. This is in
agreement with the mechanism described by Zhuang et al. (1992), who proposed an increase of iron
dissolution by the acidification of aerosol particles associated with DMS oxidation, and Johansen and
Key (2006), who showed an increase of dissolution of ferrihydrite, a proxy of iron(oxy)hydroxide found
in desert mineral dust, by photolysis of the Fe(III)-MSIA (methanesulfinic acid) complex, producing
MSA and soluble Fe. It is interesting to note that due to the high correlation between %SFe and %SAl
and %SSi, the photochemical processes could also impact the solubility of all element-bearing dust.
The possible mechanism suggested by this paper could be responsible for initiating a feedback loop
whereby the input of dust of increased solubility would result in stronger marine biogenic emissions
to the atmosphere.
In conclusion, this paper describes the very first field observations suggesting that, while airborne,
the atmospheric iron from mineral dust experiences a complex and dynamic environment where the
interplay between the input of atmospheric iron from transported dust and the marine biogenic emis-
sions from the Benguela oceanic upwelling system should be further addressed by future research.
This possible mechanism could increase the iron solubility in mineral dust, maybe also initiating a
feedback loop whereby the input of dust of increased solubility would result in stronger marine bio-
genic emissions to the atmosphere. Beside sulphur species, the role of Volatile Organic Compounds
(VOCs), in particular butene, massively emitted by the organisms in the coastal marine foam (Giorio
et al., 2022), should also be explored.

**Data availability.** Original and analysed data are available at the AERIS (https://aeroclo.aeris-
data.fr/project/, last accessed 20/07/2023). The statistical FactoMineR package is available in R (R
version 4.1.2, 2021; http://factominer.free.fr/index_fr.html, last accessed 20/07/2023). Meteorological
data from the Wlotzkasbaken station (22.31°S, 14.45°E, 73 m asl) are part of the Southern African
Science Service Centre for Climate Change and Adaptive Land Management (SASSCAL) Observa-
tionNet (https://www.sasscal.org/; last accessed 14/04/2023).



**Author contributions.** PF, DK, SJP, AN, MC, AF and SC prepared and performed the filter sampling. RT, KD, PF, SC, and CMB performed the XRF, IC and ICP analysis of the collected samples. KS and SF performed the model calculations of dust emission fluxes. JPC performed the model calculations of air mass back-trajectories. HA and JC provided with the satellite retrieval of fog and low clouds. PF, KD, RT and SJP analysed and interpreted the dataset. PF and KD wrote the paper with contributions from RT and SJP, and the remaining authors. PF and SJP provided funding. PF coordinated the research activity and supervised its planning and execution.

**Competing interests.** PF is guest editor for the ACP Special Issue "New observations and related modelling studies of the aerosol–cloud–climate system in the Southeast Atlantic and southern Africa regions". The remaining authors declare that they have no conflicts of interests.

**Special issue statement.** This article is part of the special issue "New observations and related modelling studies of the aerosol–cloud–climate system in the Southeast Atlantic and southern Africa regions (ACP/AMT inter-journal SI)". It is not associated with a conference.

**Acknowledgements.** This work receives funding by the French Centre National de la Recherche Scientifique (CNRS) and the South African National Research Foundation (NRF) through the "Groupement de Recherche Internationale Atmospheric Research in southern Africa and the Indian Ocean" (GDRI-ARSAIO) and the Project International de Coopération Scientifique (PICS) "Long-term observations of aerosol properties in Southern Africa" (contract n. 260888) as well as by the Partenariats Hubert Curien (PHC) PROTEA of the French Minister of Foreign Affairs and International Development (contract numbers 33913SF and 38255ZE). D. Klopper acknowledges the financial support of the Climatology Research Group of North-West University and the travel scholarship of the French Embassy in South Africa (internship at LISA in summer 2018). R. Torres-Sánchez acknowledges the Postdoctoral Fellowship Margarita Alsolas (University of Huelva) funded by the Ministry of Universities of Spain (NextGenerationEU). The Southern African Science Service Centre for Climate Change and Adaptive Land Management (SASSCAL) ObservationNet (https://www.sasscal.org/) is acknowledged for open-access data provision. The authors would also like to acknowledge the support by the IPGP platform PARI for HR-ICP-MS analysis. F. Lahmidi and Z. Zeng (LISA) are acknowledged for support to the ion chromatography analysis.



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
