# Peer review of "Fractional solubility of iron in mineral dust aerosols over coastal"

_EGUsphere, 2023_

## Author Comment (AC1)

**Reply to referee comments (RC) to manuscript egusphere-2023-1736'**

We thank both referees for the complete and thoughtful revision of our manuscript. In the following sections we address their comments and questions to the best of our possibilities. Replies from authors are organized by referee number. Text from referees is presented as standard text and coauthor responses are in blue.
* * *
**Referee Comments #1**

**Overview**

This manuscript presents an interesting data set on the aerosol composition and solubility of iron in Namibia. Data presented in this manuscript shows that iron solubility in the Namibian coast is rather high, compare to data collected in other deserts, close to dust sources. Authors propose that photo-reduction processes, involving methane sulfonic acid linked to marine emissions, are involved in the increase of iron solubility in the ambient air. This is a very interesting study suitable for publication in ACP. It contributes to increase our global knowledge (https://doi.org/10.3390/atmos9050201) on the variability of iron solubility in the aerosols, with specific observations at Namibia. I listed below some questions and some specific points (**SP**) that be useful to prepare the final – revised version of the manuscript for publication in ACP.

We thank Dr Rodriguez for the number of suggestions and comments which enables to enrich the data analysis and interpretation in this paper.

**Specific issues:**

**SP-01**. Line 78-82. In the introduction authors describe the role of dust deposition on the Benguela upwelling. Just to highlight that a new study may be of interest to the authors to highlight the importance of Angola and Namibian dust and the Benguela upwelling. This recent study has shown that in the North East Atlantic skipjack tuna performs northward (Jan to Aug) and southward (Sep to Dec) migrations under the Saharan Air Layer, tracking the seasonal shift of massive dust deposition; this study also indicates that the migration of skipjack tuna between Gabon (a mayor finishing area of skipjack tuna) and the area of southern Angola - Namibia (other mayor finishing area of skipjack tuna) may be modulated by dust inputs. Skipjack tuna migrates to southern Angola and Namibia in the dust season of high column dust load, this is the period when they are caught in abundance, see Fig.2A9, 2B3 and 2C3 of this is the study ( https://doi.org/10.1016/j.atmosenv.2023.120022, sorry for the self-citation, but I think it may be of interest for you). As in the case of NE Atlantic, this suggests that dust deposition (rich in Fe, P and bio essential trace elements) over upwelling waters (rich in Si and N) support zooplankton rich areas, optimal for fish larvae, molluscs, cephalopods, and large predators.

We thank the Referee #1 for indicating this publication which indeed we did not know. We have now included as a motivation in the introduction, lines 82-85: "*The inputs of Namibian (and Angola) dust in the upwelled waters could also modulate the migration of skipjack tuna between Gulf of Guinea and equatorial Atlantic, by contributing to support phytoplankton growth and hence upper trophic levels in this area (Rodriguez et al., 2023).*"

**SP-02**. A major issue in the is the extraction technique used for the determination of dissolved iron from dust aerosol samples. In this study, authors determined the concentrations of dissolved iron (by ICP-AES) by acidifying (1% nitric acid) the extraction of the sample in deionized water used for the determination of ions and cations (analysed by ion chromatography). The values of the dissolved iron concentrations, and thus iron solubility, strongly depend on the extraction technique and (specially) on the pH of the dissolution, in such a way that more acid dissolution, higher iron solubility.

Did authors measure pH of the dissolutions?

The pH of dissolution is the ones of UP water, i.e. 5.6. We omitted in the initial text to precise that after contact with UP water the aliquot were filtered on 0.2µm nuclepore filters before storage. We have added this information in the corrected text, L152 : "*The solution was filtered (Nuclepore polycarbonate filters with 0.2µm pore size) then divided into two sub-samples*". In consequence, the acidification by 1% of nitric acid happened when the dissolution is already operated and the acid adding didn't play on the dissolution process of iron.

As far as I can remember now, there are, at least, three broad-group of extraction techniques:

1) the one used by the authors (I suggest to include references to other studies which have used the same technique), where the pH of the dissolution may be influenced by the acidification technique used and the presence of ionic balance between acidic species and basic species, including the buffering capacity of buffering capacity of $CaCO_3$.

2) the extraction of the sample in ammonium acetate leach at pH 4.7, e.g. as used by Baker and Jickells (2017) (http://dx.doi.org/10.1016/j.pocean.2016.10.002)

3) solid phase techniques extraction in real sea water (at the pH of the real sea water, pH ~ 8.1), e.g. as used by Rodríguez et al. (2021, https://doi.org/10.1016/j.atmosenv.2020.118092) and Ravelo et al. (2016, http://dx.doi.org/10.1016/j.atmosenv.2016.03.030); The use of real seawater allows considering the potential role of the organic ligands present in the ocean, which complex Fe to keep it in solution in excess of its solubility.

In principle, each extraction technique should be a proxy of an atmospheric process. Thus, iron solubility determined in deionized water would be a proxy of iron dissolution by in-cloud processes, often followed by wet dust deposition, whereas iron solubility determined in sea water would be a proxy of dry dust deposition in the ocean. In general, iron solubility in deionized water tends to be higher than in sea water (due to the lower pH of the former, 6 vs 8). In the manuscript, authors did a too general (vague) comparison with other studies, both in the section 3.2 and in the abstract (lines 43-45) ;

According to section 3.2 and table 2, iron solubility during dust events is 7.9% (6.9% if removing 1 specific event, line 287), a value much higher than that observed in the dust aerosol samples from the Sahara, which is typically from 0.5 to 0.7 % in real sea water, see begging of section-3 of Rodríguez et al. (2021, https://doi.org/10.1016/j.atmosenv.2020.118092) (sorry for the self-citation again, but just to point that there is a summary and discussion on this topic with many references in this paper).

I suggest to authors to include a brief text (may be in the methodology and sections 3.2) describing that iron solubility will depend on the extraction technique, then authors may

explicitly say that they compare their results with previous studies which have actually used the same technique (they may even compare with studies based on sea water extraction, for which lower solubilities are expected).

Indeed, the solubilities values are very dependent on the extraction techniques. We have now included the explanation of the used leaching protocol and the consequences of this choice on the comparison with the other studies in P7-L174: "*Here, a leaching protocol using ultrapure water (UPW) was used to simulate wet deposition of particles, since the wet deposition dominates the total iron supply in the Southern Atlantic Ocean (Chance et al., 2015). Moreover, the UPW leach enables the chemical reaction between iron with organic or inorganic ligands, naturally dissolved from the particulate aerosols into rain droplets. However, it is known that the extraction protocol modulates dissolution process and hence the values of iron fractional solubility, in particular the estimates using UPW are higher in comparison to these one using seawater, but lower than the acidic, buffered or reduction agent leach (Perron et al., 2020).*"

In complement, we have also specified the conditions of leaching when we compare our data with the literature L313 in the discussion section: "*For both conditions, the observed range of variability is high and consistent with previous observations over the Southern Atlantic Ocean (2.4-20 %, Baker et al., 2013; 1.3-22 %, Chance et al., 2015), as well as over the Southern Indian Ocean (0.76-27 %, Gao et al., 2013), using acetate buffer leach at pH 4.7 (0.4μm) which extracted 1.4 times more Fe than UPW protocol (Perron et al., 2020).*"

We haven't added the comparison proposed by the referee with Fe solubility in Saharan dust using seawater leach over Northern Atlantic Ocean, since we estimated it is not relevant here due to the large difference of conditions.

**SP-03**. This is just a suggestion. Equation 1 is used for estimate dust mass concentration. In this approach elements are assumed to be present as oxides, a hypothesis which is not actually true (since most of the mineral dust are Si and Al aluminosilicate minerals, clays), but nonetheless this is a approach widely used. Why is the factor 1.12 used?, is it to compensate the average contribution of any element (e.g. that aluminium accounts for 8% of the dust mass)? If so, authors could verify it or simply determine the real one value the scatter plots of Al or Si vs gravimetric $PM_{10}$ (is available) in ochre dust samples.

This is important because in line 273-274 it is stated that total iron accounts for 5.8% of dust (total iron / EDM). This value is somewhat higher than that in Saharan dust, which is within 3.9 - 4.0% with an Fe/Al ratio = 0.5, which is lower than the 0.76 found by the authors in Namibia (so there is a data consistency). If authors have gravimetric $PM_{10}$ concentrations and their samples are by far dominated by dust (ochre dust colour, as Fig.1B of the study https://doi.org/10.1016/j.atmosenv.2019.117186 ) they could do the scatter plot of Al versus $PM_{10}$ and determine the actual contribution of Al (similar to Fig. 1A of the study Rodriguez et al., 2020, https://doi.org/10.1016/j.atmosenv.2019.117186 ), then, the EDM would be:

EDM $(mg/m^3)$ = (1 / slope (Al vs $PM_{10}$)) · Al $(mg/m^3)$          Eq-R1

Authors could also do it with silicon (which actually much better)

EDM $(mg/m^3)$ = (1 / slope (Si vs $PM_{10}$)) · Si $(mg/m^3)$          Eq-R2

In the Sahara slope (Al vs $PM_{10}$) = 0.079, i.e. Al accounts for 8% of dust

In the Sahara slope (Si vs $PM_{10}$) = 0.16, i.e. Si accounts for 16% of dust

According to my experience with Saharan dust samples, equation 1 may underestimate EDM by a 10%, compared to the Eq R1 and R2. Just to remind that this (SP-02) is just a suggestion, in case authors find it interesting.

Referee #1 is correct; the evaluation of the total dust mass is a crucial point. Unfortunately, the gravimetric measurement of the PM10 mass concentrations was not possible, which obliged us to using alternative ways to estimate the dust mass concentration from elemental analysis. Estimate of mineral dust concentration has been made by combining the concentrations for crustal elements, multiplied by the respective molar correction factors derived assuming that the elements are present in the form of oxides, according to the composition of the Earth's crust given by Lide [1992]. Finally, molar correction factors have been proportionally increased by 12% to account other compounds present in the average sediment.

This formulation agrees within 5% with the dust mass concentration estimated from elemental aluminum considering it that accounts for 8.31% of the total dust mass (shown here below). Note that 5% is within the XRF analysis analytical error, so that the difference is not significant.

[Figure]

**SP-04**. Lines 148-155.

Fluorine is cited in the abstract, but not included here (methodology section). How was fluorine analysed? It should be described.

Fluorine was analysed by ion chromatography method. It was omitted from the list of compounds analysed by this method in the methodology section. It now appears (L156).

**SP-05**. Lines 195-206. Was the presence of fog verified with local in-situ measurements of local meteorological data of relative humidity (RH)? (table 1?), if these data are available, authors cloud just flag their data and compare them. Also, to validate their method.

The fog record is actually available as part of the Wlotzkasbaken meteorological dataset. The suggestion of Referee #1 is good and actually we have checked the good correspondence between the fog model, the RH and fog flag in the meteorological data.

**SP-06**, lines 202: < The FLC product does not specifically distinguish between fog and low clouds >. To use local meteorological data of RH could help to distinguish fog from clouds.

Indeed Referee#1 is right when that the local relative humidity could be used to provide further elements of distinction between low level clouds and fog, and indeed values of relative humidity are available as reported in Table 1. However, what we are focusing on are really the large regional features along the pathway of transport of mineral dust. In this respect, the satellite-based FLC product is more adapted, albeit, we do agree, perfectible.

**SP-07**. Lines 237-241. This is very interesting. I suggest to include into the body of the article (not in the supplement) a figure with some back trajectories over a satellite view (e.g. Google Earth) of the regions; it would be useful for reader that do not know the region. The number of events is rather low, so a composite could be done with this.

The back-trajectories for dust events have added in Figure 2 in the principal text. We have decided to keep the back-trajectories over dust AOD and wind maps, as in supplement S1, to provide the information both on the active dust source and air mass transport.

**SP-08**. Lines 242-246. The formation of fog is typical of the coast of subtropical deserts characterised by upwelling of deep cool waters. Trade winds plays a key role in prompting such upwelling (they are the actual prompters of the southern current and Benguela upwelling) so it should explicitly be cited.

The text of the paper now mentions trade winds as requested.

**SP-09**. Section 3.2 presents the data of soluble iron, aluminium and silicone. Results are presented in Table 2 (comparing dust versus background conditions), then temporal evolution of SFe is shown in Fig. 2 (segregating dust from background conditions) and finally Fig 3 shows the plots of total Fe-vs-total Al, total Fe-vs-total Si, dissolved Fe-vs-total Al, dissolved Fe-vs-total Si. Iron solubility (%S) under dust and background conditions, values are very close, 7.9 and 6.8, respectively. According to line 287, if the dust event 11 is not considered, then iron solubility is similar under dust (6.9%) and background conditions (6.8%). Authors conclude that Fe and DFe have a (line 321).

In my modest opinion the data analysis is rather short and not conclusive. It seems that under background condition there is also a significant amount of dust (probably up to exceeding 10 mg/m$^3$, according to Fig.3) and that it is a source of iron. How was background conditions defined? We based on Klopper analysis, integrating Al and nss-Ca2+ content as mentioned in L217: A previous study in this site by the same group (Klopper et al., 2020; Atmos. Chem. Phys., 20, 15811–15833, 2020, https://doi.org/10.5194/acp-20-15811-2020 ) found very interesting results that may be use useful for the interpretation of the soluble iron data, they found that: 1) main sources of aerosols: sea salt, mineral dust, fugitive dust, industry and ammonium neutralized, 2) As, Zn, Cu, Ni and Sr attributed to combustion of heavy oils in ships, and 3) V, Cd, Pb and Nd of fugitive emissions from mining actives.

Is there a fraction of soluble iron linked to fugitive mining dust?, Is it contributing to the background? Authors could use the results of Klopper et al. to identify if dissolved iron is linked to any of these sources and during background and even dust conditions. Even if most of total iron may be linked to dust, an important fraction of dissolved iron may be linked to other sources. Have authors tried a source apportionment of soluble iron?

Authors could do a PMF as Klopper et al. (2020) or simply use the knowledge obtained in the study of Klopper et al. to apply the method used by Rodriguez et al. (2021, https://doi.org/10.1016/j.atmosenv.2020.118092) by scatter plots of soluble Fe to total Fe and soluble Fe to Ni/Al (Rodriguez used V, but authors may use Ni as tracer of heavy fuel oil combustion since V is linked to primary mining activities in their study site, according to Klopper et al., 2020). How does the plot of iron solubility SFe (%) versus Ni/Al ratio looks?

It could serve to know if fuel oil combustion is contributing to soluble iron or not. Ni is a tracer of heavy fuel oil combustion, which is a source of soluble iron due to the emissions/formation of ferric sulphate and nanocrystals of magnetite aggregates (Fu et al., 2012; https://doi.org/10.1021/es302558m ), formed at temperatures >800 ∘C, followed by sulphuric acid condensation (Sippula et al., 2009 https://doi.org/10.1016/j.atmosenv.2009.07.022 ). According to Klopper et al. (2020) this region is affected by the emissions of ships in the Cape of Good Hope and these emissions may also be impacting during dust events since southern winds prevailed at the sampling site during the dust episodes (according to wind direction in table 1). This may help in the data analysis. Other data analysis that may help to enrich data treatment (just suggestions):

Has the plot SFe (%) vs Fe a hyperbolic trend?

To include (as supplement or at least cite how they look) the plots of (1) dissolved iron to total iron, (2) dissolved iron to Ni/Al and (3) dissolved iron to sulphate/Al, nitrate/Al and fluoride/Al may help to understand the behaviour of the data and sources of soluble iron.

We thank Dr Rodriguez for the number of suggestions and indications for further data processing, and the publication. We acknowledge the fact that section 3.2 introducing the Fe solubility is too concise and misses some important figures. Note that we chose another representation, the PCA, to illustrate the results, but indeed there is a lot to learn from a close look to correlations and scatter plots.

First of all, the background conditions were defined in Klopper et al. (2020) based on the time series of the mass concentrations of elemental Al and nss-$Ca^{2+}$ (reported in L228). Events of mineral dust were identified as time at which the mass concentrations exceed a 10-days moving average for a significant amount of time. We acknowledge that this classification could miss some short term episodes of very local emission or transport.

The suggested figures are reproduced here below (we additionally show the correlation of SFe% with the ratio nss-SO4/Al). These figures have been already studied, but we estimated that they are not conclusive to be presented in the manuscript. In our opinion, they confirm the fact that the attribution of sources and drivers of the Fe solubility in Henties Bay is complicated.

The soluble-to-total Fe values both in dust and background conditions do not show a hyperbolic trend (see Figure below). The hyperbolic trend of Fe-dust solubility increases with the decrease of dust loading (Fe content) is currently observed in the literature and the cause of this pattern is always debated. A first explanation is that the increase of solubility is due to a mixing with more soluble anthropogenic iron, and the second one is linked to the atmospheric processing of dust during transport. The observation of a such trend should be useful for data analysis. However, the observation of hyperbolic pattern demands to have a dataset with contrasted iron content (or dust concentrations, either due to different origin of air masses (e.g. Baker et al., 2006: https://doi.org/10.1016/j.marchem.2005.06.004 or Baker at al., 2020:

https://doi.org/10.1029/2019GB006510) or due to the evolution of dust concentration with long range transport as in Rodrigez et al. (2021). Namibian boundary layer is a complex atmosphere impacted by different emissions and characterized by very peculiar atmospheric conditions, relative humidity in particular, as already shown by Klopper et al (2020), and as discussed in the paper. This is significantly different from the situation described in Rodriguez et al. (2021) for northern African dust, which is transported in the Caribbean in a stable and thermodynamically confined layer (the SAL) in the free troposphere, only subsiding to the surface after a significant amount of time. We attribute the absence of hyperbolic trend to the fact the aerosol in the Henties Bay atmosphere is mixed, even during dust episodes, so well-homogenized. This as a result of the fact that transport of dust occurs exclusively at low altitudes within the boundary layer, which is thermodynamically isolated from the free troposphere. This homogenization effect is augmented by the limited range of iron content in our dust samples (0.07 to 2.8 $\mu g.m^{-3}$) in comparison to background samples (0.02 to 1.2 $\mu g.m^{-3}$).

[Figure]

[Figure]

As for dissolved/soluble-to-total iron, no plot with typical anthropogenic markers was effective tool to emphasize a potential role of anthropogenic iron on values of solubilities. The correlation between iron solubility and the ratios of SO4, nss-SO4, and NO3 to Al are mostly insignificant for dust conditions, while tends to show some relationship in background conditions. The correlations are more pronounced with respect to the F/Al ratio, both in dust and background conditions, as a result of the specificity of the Namibian soils rich in fluorspar minerals, as mentioned in the paper.

[Figure]

Moreover, PCA results confirm that no trend of evolution between %SFe and interelement ratio, used as source marker:

[Figure]

Variables - PCA

**SP-10.** The average iron solubility that authors found in Namibia, close to sources, is much higher than in other sites close to dust sources. I think that this is something that should be explicitly say in the section of results, but also in the abstract and in the conclusive remarks. Average SFe during dust events found in several studies, = 6.9% in Namibia, 0.7% in Tenerife (close to Sahara), 1.3% in Barbados (distant to sources). Many studies have found SFe(%) < 1% during dust events and then increase up to 10% along several days of aerosol aging.

The increase of %SFe during Saharan dust transport is known (e.g. Longo et al., 2016). Yet, it is not demonstrated for Southern dust sources. Here, the comparison with the previous observations after Namibian dust transport shows that the range of %SFe values after transport were close to our ranges, although they were obtained with a protocol optimizing the extraction (as mentioned in L313). That means that our %SFe values could be likely lower than in the literature, however we have no way of knowing for sure. It is the reason why we cannot be conclusive on this point and that we have just compare our data with measurements over the Southern Oceans, under influence of Namibian sources (L313). However, in order to specify this aspect, we have added L315: ". using acetate buffer leach at pH 4.7 (0.4µm) which can extract 1.4 times more Fe than UPW protocol (Perron et al., 2020)."

**SP-11**. It would be interesting to label with A, B, C and D each plot of Fig 3 and make the proper reference to them in the text.

This is now done

**SP-12**. According to line 287, if the dust event 11 is not considered, then iron solubility is similar under dust (6.9%) and background conditions (6.8%). This is a very distinctive feature that should explicitly be compared with other studies. North African dust observations in the Atlantic have shown that iron solubility is lower during dust events (high dust concentrations) than in the background aerosol (dust at low concentrations), see as example: Baker and Jickell 2006 and Rodriguez et al. (2021) both included in the article.

As previously mentionned, we attribute this to efficient ageing by marine biogenic emissions more than source.

**SP-13**. Section 4 Discussion. This section also includes a presentation of results with plots, which is something that usually goes into the section of results. Authors may consider to merge both sections as a single section Results and Discussion.

We understand Referee #1 concerns, and as a matter of fact, we struggle in finding the right structure to this paper. However, we believe that separating results and discussion allows us to distinguish between what is linked to the data itself, and the interpretations we make of it.

**SP-14**. Figure 5. It would be useful to put labels A (for FS(Fe)), B (Formate), C, D, E, F and G (nssSO4) and cited them in the text.

We privilege this representation as it is more straightforward for the reader who immediately can see which compound the figure refers to. With the permission of Referee #1, we would prefer leave the labelling as it is.

**SP-15**. Figure 5. Caption needs to cite that the first plot is dimensionless and not in $mg/m^3$. Authors may also consider to put the first plot as %.

This is now corrected

**SP-16**. Result describe in Lines 407-408 is very interesting. How does the plot %SFe vs MSA look?

Added in supplement (Fig. S54) with also relation between %SFe and all the secondary compounds concentrations.

**SP-17**. Section 4.2 is very clear, however section 4.3 is a little bit farragoes, it would be interesting to smooth and shorten the text.

This section was rewritten to improve clarity. It now reads "*Formenti et al. (2018) have shown that in the Austral winter, when the synoptic circulation is dominantly anti-cyclonic, air masses laden with light-absorbing aerosols either from ship pollution or biomass burning can be transported to HBAO (Formenti et al., 2018). However, the lowest Fe solubility (< 5%) was measured in July and August 2017, and no correlation between the %SFe and the percent mass fraction of iron from sources other than dust can be found in our data. The mass apportionment of iron reported by KL20 indicates that, during the dust events and the background periods, respectively, 7% and 29% of the mass of total elemental Fe was not associated to mineral dust, but rather to a factor indicated as "ammonium-neutralised component", mostly characterised by secondary species, and non-sea-salt potassium (nss-K+). The PMF analysis indicated that the "ammonium-neutralised component" was associated to photo-oxidation of marine biogenic emission but also episodically to biomass burning. This component includes oxalate, the most*

*concentrated organic species at HBAO, and the strongest of the organic ligands promoting the photo-reduction of iron in mineral dust, henceforth the increase of its fractional solubility (Paris and Desboeufs, 2013). Surprisingly, excepted individual cases (Dust 13), our analysis does not show this strong link (Fig. 6), and indeed, contrary to the SFe%, the oxalate concentrations measured at HBAO was practically constant with time. The possible pathways of oxalate formation in this complex atmosphere are numerous through the year, from natural and anthropogenic sources (marine, heavy-oil combustion, biomass burning) and in-cloud and photo-oxidative processes (Baboukas et al., 2000; Myriokefalitakis et al., 2011)."*

**SP-18**. Conclusive remarks could be summarised, just focusing on the most relevant findings and the ideas proposed. I think that a sentence saying that average iron solubility in Namibia, close to sources, is (6.9%) much higher than in other sites close to dust sources (<1%). Then propose the photo-reduction processes, involving methane sulfonic acid linked to marine emissions, as potential process favouring such high iron solubility.

These suggestions were accepted. The concluding remarks section was shortened and rewritten.
* * *
**Referee Comments #2**

Desboeufs et al. present a novel dataset of Fe solubilty from aerosols collected from Namibia, an understudied, but regionally important and poorly characterised, dust source. It was interesting to see the apparent seasonal variability and possible links with biogenic emissions from the Benguela EBUS.

We thank Dr Shelley for the number of suggestions and comments which enables to enrich the data analysis and interpretation in this paper.

A small detail that is missing is the acknowledgement that differences in leaching methods also leads to some variations in aerosol Fe solubility, so the best we can hope for when comparing data from aerosols of broadly the same source using different techniques is that the values are broadly consistent (so, trends rather than absolute values is what we are looking at). It would be invaluable to future studies and comparisons, particularly those with a modelling component, if more details of the leaching method are included.

Indeed, the solubilities values are very dependent on the extraction techniques. We have now included the explanation of the used leaching protocol and the consequences of this choice on the comparison with the other studies in P7-L175: "*Here, a leaching protocol using ultrapure water (UPW) was used to simulate wet deposition of parti-cles, since the wet deposition dominates the total iron supply in the Southern Atlantic Ocean (Chance et al., 2015). Moreover, the UPW leach enables the chemical reaction with organic or inorganic ligands, naturally dissolved from the particulate aerosols. However, it is known that the extraction protocol modulates dissolution process and hence the values of iron fractional solubility, in particular the estimates using UPW are higher in comparison to these one using seawater, but lower than the acidic, buffered or reduction agent leach (Perron et al., 2020).*"

In complement, we have also specified the conditions of leaching when we compare our data with the literature L310 in the discussion section: "*For both conditions, the observed range of variability is high and consistent with previous observations over the Southern Atlantic Ocean*

*(2.4-20 %, Baker et al., 2013; 1.3-22 %, Chance et al., 2015) and the Southern Indian Ocean (0.76-27 %, Gao et al., 2013), using acetate buffer leach at pH 4.7 (0.4µm) which can extract 1.4 times more Fe than UPW protocol (Perron et al., 2020).”*

I would also like to see more in depth analysis of the data, for instance, discussion on whether correlations are statistically significant.

We re-worked all the paragraph presenting PCA analysis, including statistical quantification (see detailed reply in the suggestions on L428 about this point).

Other than that, I have made some minor suggestions for clarity, detailed below.

Line 31. Insert growth before limiting: Done

Line 71. I think there is an extra ) after Etosha Pan which needs removing: Removed

Line 75. If you include the Southern Ocean in the list of where southern African dust can be transported to, the next sentence will be better linked.

Changed

Line 94. In the Methods you say that elemental determination was by WD-XRF not ICP(-MS?). I have re-read the Methods section and now see that it was total concentrations that were analysed by WD-XRF and soluble elements either by ICP-AES or ICP-MS. You could either clarify here that different methods were used for total and soluble elements, or just end the sentence at metals.

Done

Line 101. As you are determining MSA from MQ water leaches, the MSA that you detected was water soluble rather than particulate, so it might be best to remove the word particulate from in front of MSA.

Done

Line 103. Replace dust with Fe here: Done

Section 2.2. What was the rationale for collecting different day/might samples and the 3 h gap between sample collection? Do you know if any large dust events or other unusual conditions were missed by sampling non-consecutive weeks?

The rational for the 3-hour gap was to confine sampling to daytime (with solar radiation) and night-time (without solar radiation), regardless of time of the year. Regarding the continuity of measurements and the possibility that dust events other than those sampled occurred, we cannot exclude but unfortunately we do not have a continuous records of complementary measurements (our nephelometer experienced problems). Satellite data are not reliable because of the cloud cover.

Line 135. Does the 176 samples include the 13 blanks? Were 163 samples + 13 blanks or 176 samples + 13 blanks collected?

We have changed: "176 samples (+ 13 blanks, one per week of sampling)"

Line 149. Missing citation for the aerosol leach. Was the leachate filtered? If so, through what pore size? This is important information for future comparisons with the data set and needs to be included.

Added

Line 260. Enhances: Done

Line 273. It might be worth noting that this is greater than the 3.5% in UCC (Taylor and McLennan, 1995) and closer to the value of 5.04% in Rudnick and Gao (2003), and that this could be due to natural differences in elemental abundance between UCC and Namibian source material and/or that UCC is slightly imperfect proxy for aerosol dust.

We agree with Referee #2 on the fact that it is interesting to compare the Fe abundance found in our samples and the usual natural abundance in upper crust. We added a sentence to introducing this comparison in L291: "*These values are quite superior to usual Fe content recommended in upper continental crust models (3.5% for Taylor and McLennan or 5.04 +/- 0.53 % Rudnick and gao, 2004) and estimated in Saharan dust (4.45% for Guieu et al., 2002; 4.3 to 6.1% for Lafon et al., 2006 or 4.5% for Formenti et al., 2008). Keeping in mind that this Fe abundance is estimated, this suggests that the Namibian aerosol dust could be enriched in iron in comparison to upper crust and dust provided by Saharan sources.*"

Line 283. %SFe is higher during dust events but not significantly so due to the large variability. And only by ~1% so it is hard to accept that that there is a real difference between dust and background in this data, especially as you state that it is the data from Dust 11 that skews the data. Perhaps some rewording needed to make it clearer that there is no difference between the %SFe in dusty/background samples. Perhaps swapping the last two paragraphs starting in line 282 and 291 around would help make this clearer. This similarity in dusty and background %SFe is noteworthy as it contrasts with other regions under the influence of episodic dust events, which should be mentioned.

We apologize for the confusion as this is clearly stated in lines 305 and L309: "*The uniformity of iron solubility values between background and dust periods contrasted with the observations made in regions where the dust influence is sporadic and the origin of Fe is associated to various sources (e.g. Shelley et al., 2018). That is consistent with a main dust source of iron in our samples, as indicated in KL20.*".

Line 289. The fractional solubilities from the Baker and Gao studies both used ammonium acetate leaches, whereas, your study used MQ. In addition to mineralogy influencing fractional solubility, several recent studies (and now there is a SCOR Working Group is looking at this topic) have compared the use of different leaching schemes on trace element fractional solubility and concluded that caution should be used when comparing results from leaches using different leach media as the chemical composition of the leach media influences the amount of X that dissolves (from which fractional solubility is calculated, e.g., Perron et al. 2020). The point being your soluble concentrations and %SFe are likely lower than those of Baker and Gao (although we have no way of knowing for sure), supporting the argument that %SFe increases during atmospheric transport (e.g., Longo et al., 2016), while still producing results which are broadly consistent with the cited studies. While not suggesting this is the place for a discussion

on leaching schemes, it is important to give as much detail as possible about the leach used in this study in the Methods to allow better comparisons between this dataset and others in the future. Therefore, you should mention the differences in leach media (and the impact on (fractional solubility) here.

See detailed reply in the general comments in the beginning of the review

Fig. 3. The equation in grey is very hard to read. Change to black. Done

Line 330. Differences in leaching methods also leads to some variations in aerosol Fe solubility, so the best we can hope for when comparing data from aerosols of broadly the same source using different techniques is that the values are consistent. The exciting finding is that although there was no difference in solubility between dusty and background samples in this study, except for Dust 11 and 13, there was a seasonality.

Fig. 4. The light yellow dots for Dust 4 are very difficult to see. Could a different colour be used?

Done with dark color for all the points

Line 384. Soluble fraction: Done

Line 397. As you were you determining MSA that has dissolved from particulates rather than particulate MSA, best to remove particulate, perhaps? Done

Line 401. This should be part of the previous paragraph: Done

Fig. 5. Light pink is hard to see. Could you switch the order so that MSA is under the %SFe plot? Done

Line 420. 'the closer they are in the circle': Done

Line 428. Does the PCA indicate which relationships are significant and which are not? It would be useful to have this information.

No high correlation was found between %SFe and the different measured parameters (meteorological data, secondary compound concentrations, interelement ratio, PMF results..), as presented in supplement material (Fig. S4) and in the reply to referee #1, except with %SAl and %SSi. We have added this information L 446. The PCA was used to explore the time trend between the different parameters. In order to quantify the contribution of variable in the PCA, we used the squared cosine (cos2), which gives the importance of a component for a given, observation. Thus, the closer a variable is to the circle of correlations, the better its representation on the factor map (and the more important it is to interpret these components). We have considered that the contribution was not representative when the cos2<0.4. In order to clarify these points, we have changed the PCA plot (Fig. 7) by representing with a same colour the variables with a same trend (i.e. groups in the PCA plot) and we mentioned in the figure caption "cos2<0.4" as limit to estimate the existing trend between variables.

L455: "*Fig 7. PCA analysis performed from the database including %SX, TX, EDM, secondary ions concentrations and meteorological parameters. The colour of variables by groups is defined by a clustering algorithm, tending to find clusters of comparable spatial extent. Each*

*colour corresponds to a cluster of parameters which evolve in the same way. Formate, nitrate, ammonium, acetate, humidity, and wind speed are not visible in the plot showing that they are not significantly correlated with the other parameters ( i.e. their squared cosine < 0.4)."*

Moreover, we have also clarified the text by indicating the different groups found by PCA: *"The PCA correlation plot (Fig. 7) emphasizes 3 groups of dependent parameters: 1. a high correlation between total Fe, Al and Si concentrations and dust loading (EDM), as previously identified in Table 1 and Fig. 3, 2. a relation between oxalate and nss-SO42- concentrations, suggesting a common chemical process of formation, and 3. the dependence between %SFe (%SAl and %SSi), the MSA concentrations, the solar irradiance and to a lesser extent with. the wind speed. While it is expected that the emission of mineral dust occurs when the wind speed is high, the correlation of %SFe with wind speed is rather surprising as both Table 1 and Fig.2 show that since the %SFe is independent of the dust load.(Fig. 2 and 6). Fig. S4 in the supplementary material present the plots between %SFe, MSA concentrations, solar irradiance and wind speed for background and dust events. The correlation between the wind speed and the MSA concentrations (Fig. S4) is consistent with Andreae et al. (1995), who demonstrated how, in this area due to persistent phytoplankton bloom, the atmospheric concentrations of dimethylsulphide (DMS), the gaseous precursors of MSA, depend on the sea-to-air flux, in turn is determined by the concentrations in the ocean water and the surface wind speed."*

Line 466. Very long sentence. changed

Line 468. Include the range and median value. done

Line 473. Concluding remarks: done

Line 484. 'average water-soluble Fe fractional solubility is…' done

Line 489. What do you mean by a benchmark lab experiment? + Line 490. I think some values are needed here for the soil Fe fractional solubility. This perhaps highlights the fact that soils aerosolised in the lab are not the exactly the same as the aerosols collected on filters.

The sentence was meant to introduce experiments which were done in our simulation chamber where we use a shaking system to produce aerosol dust from real soils. To improve clarity, it has been rewritten as *"The measured iron fractional solubility is significantly higher than obtained from dissolution experiments, with an identical protocol, of mineral dust aerosol samples collected on filters after laboratory generation from the same soil (< 1%; Formenti et al., in preparation, 2024)"*. Note that accordingly to the request of Referee #1 this sentence is now moved to section 3.2, lines 293-296.

Line 496. Replace conversely with however and continue from previous sentence (which should not be a stand-alone paragraph).

We choose to move "conversely" since the previous paragraph was removed.

Line 501. Namibia: Done

Line 507. Perhaps include same before photochemical Done

Line 509. 'increased trace and major element solubility': Done

Line 511. Remove very: Done

 It is redundant if this is the first dataset of its kind. Perhaps it is also worth stating again that the conditions in the MBL result in deliquescent aerosols at the study site.

This is now done.

---

## Author Response (AR3)

**Reply to editor's comments:**

We would like to express our gratitude to the editor for his thorough review. As requested, we have revised and condensed the abstract to meet the 250-word limit. We greatly appreciate the technical revisions and have incorporated most of the suggested minor changes. However, there are two points where we deviated from the suggestions:

- Line 104: We believe that 'highest' accurately describes the concentrations, so we maintained the phrase as 'these highest concentrations.'
- Lines 279 and 406: We prefer to use the integral form, so we kept 'fractional solubility of Fe, Al, and Si' as is, but added the percentage form as '%SFe, %SAl, and %SSi' for clarity.

Regarding the figures, all modifications have been implemented. Figure 3 now has an improved resolution, although we acknowledge concerns related to the pasting process. The original figure provided as an individual file is in high resolution.

For figure citations in the text, we have followed the ACP guidelines for authors, using 'Fig.' when it appears in running text and 'Figure' at the beginning of a sentence. Bold text is reserved for figure captions.

Since the last revision, a DOI has been assigned to the dataset. In the 'Data Availability' section (L548), we have included this information: 'Atmospheric concentrations of total and dissolved elements and water-soluble ions measured over coastal Namibia in 2017 are available in Easy Data (Formenti et al., 2023, https://doi.org/10.57932/2ac79cd1-282a-4004-87d5-38f0ebcaf40c).'